# Longitudinal Shifts in Movement Behaviors during the COVID-19 Pandemic: Relations to Posttraumatic Stress Disorder among University Students

**DOI:** 10.3390/ijerph192013449

**Published:** 2022-10-18

**Authors:** Jie Feng, Patrick Wing Chung Lau, Lei Shi, Wendy Yajun Huang

**Affiliations:** 1Department of Sport, Physical Education and Health, Hong Kong Baptist University, Kowloon Tong, Hong Kong, China; 2Laboratory of Exercise Science and Health, BNU-HKBU United International College, Zhuhai 519088, China

**Keywords:** physical activity, sedentary behavior, sleep, posttraumatic stress disorder, undergraduate, COVID-19

## Abstract

This study examined the longitudinal changes of movement behaviors and their relationships with posttraumatic stress disorder (PTSD) among university students during the coronavirus disease 2019 in China. A total of 569 university students completed online surveys twice (Time 1: between December 2020 and January 2021; Time 2: between November and December 2021). Physical activity, sedentary behavior, sleep duration and quality, as well as PTSD were self-reported. According to Canadian 24-h movement guidelines, the longitudinal shifts in each movement behavior from Time 1 to Time 2 were divided into four categories (consistently meeting the guidelines, from meeting to not meeting the guidelines, from not meeting to meeting the guidelines, and consistently not meeting the guidelines). Generalized linear mixed models were conducted using 410 valid responses (20.2 ± 1.0 years old at Time 2, 41.2% males). From Time 1 to Time 2, 22.2%, 2.0%, and 45.6% of the students consistently met the physical activity, sedentary behavior, and sleep guidelines, respectively. Compared to those who consistently met the sedentary behavior guideline, students who consistently failed to meet or changed from meeting to not meeting the guidelines had higher levels of PTSD. Students who had poor sleep quality at both time points or changed from good to bad sleep quality had higher levels of PTSD than those who maintained good sleep quality over time. Compared to those who consistently failed to meet the guideline, students who consistently met the PA guideline had higher levels of PTSD. These findings highlight the needs to improve and maintain healthy behaviors for mental health.

## 1. Introduction

The university years occur when late adolescents are transferring to adulthood; students at this age may establish new habits of movement behaviors. Movement behaviors, including physical activity (PA) [1], sedentary behavior (SB) [2], and sleep [3] in isolation, have been found to be closely associated with a variety of health outcomes. A healthy combination of movement behaviors (high PA, low SB, sufficient sleep) was associated with greater health benefits among adults [4,5]. However, previous studies have documented unhealthy movement behaviors among university students, such as insufficient PA [6], overmuch SB [6], insufficient sleep duration [7], and poor sleep quality [7]. After the outbreak of the coronavirus disease 2019 (COVID-19), many measures were taken to decrease the spread of the virus, such as stay-at-home requirements, travel restrictions, and facial covering [8]. Additionally, more than 150 countries adopted school closures [8]. Not surprisingly, in response to the disturbed routines, unfavorable changes of movement behaviors, such as decreased PA of all intensities [9], higher level of SB [10], delayed bedtime and wake up time [11], and worse sleep quality [12], have been widely reported among university students.

The rapid transmission and mortality risk of the virus, the subsequent social distancing, increased financial burden, resources shortage, decreased basic services, and decreased social support could negatively impact mental health [13,14]. A systematic review reported a high prevalence of mental health problems (e.g., stress, anxiety, depression, posttraumatic stress disorder [PTSD]) among the general population after the pandemic outbreak [15]. Among university students, a systematic review covering 706,415 participants found more than ten mental health problems, with depression, anxiety, stress, and PTSD being the most prevalent [16]. Among them, PTSD, in particular, persists for a long period even after the epidemic has vanished [17]. PTSD often developed after people experienced a traumatic event, and is characterized by avoidance to the event, negative changes in mood and cognition, reexperiencing the event, hyperarousal, etc. [18]. Individuals with PTSD are at a higher risk of experiencing cardiovascular disease [19], all-cause dementia [20], and even suicides [21]. A cross-country study observed PTSD symptoms in response to pandemic-related direct (e.g., exposure to the virus) and indirect (e.g., unemployment, social distancing, increased caring burden, changes in daily routine) events [22]. With regard to university students, a systematic review pooling 90,879 university students from 15 countries reported that 30% of them had PTSD during the COVID-19 pandemic (from December 2019 to July 2020) [23]. This further shows the importance of the early prevention and treatment of PTSD.

It is noteworthy that movement behaviors (e.g., PA, SB, sleep) were closely associated with PTSD [24,25]. More specifically, healthy adults with higher PA had lower levels of PTSD [26], whereas longer sedentary time predicted more symptoms of PTSD [24]. Regarding sleep, limited sleep time, low sleep quality, and PTSD were very likely to co-concur [27]. Such cross-sectional associations have been echoed by research findings observed during the COVID-19 pandemic. For example, regular and sufficient PA after the pandemic outbreak was associated with lower levels of mental health problems (e.g., depression, anxiety, stress) among university students in China [28]. Different findings were observed in our previous study among Chinese university students, that is, higher levels of PA during the pandemic peak (January to March, 2020) was associated with a higher risk of PTSD [29]. Another study among more than ten thousand university students observed positive associations between screen time and depressive symptoms [30]. Moreover, our previous study conducted among Chinese university students found that longer sleep duration and better sleep quality were associated with lower levels of PTSD one year after the outbreak of the pandemic [29].

Though a wide range of studies have compared movement behaviors before and during the early phase of the COVID-19 pandemic outbreak, there is scarce evidence on how movement behavior shifts across time predict PTSD. Therefore, this longitudinal study aimed to investigate the relationships between the changes of movement behaviors at two time points (one year and two years after the epidemic outbreak) and PTSD. Specifically, at Time 1 (between December 2020 and January 2021, one year after the pandemic outbreak) and Time 2 (between November and December 2021, two years after the pandemic outbreak), universities have reopened and daily routine has become normal in China. To classify the longitudinal changes of movement behaviors, the Canadian 24-h movement guidelines for adults were applied [31], which integrate all three movement behaviors across a 24-h day. Based on the PA, SB, and sleep recommendations of the Canadian 24-h movement guidelines for adults [31], the longitudinal shifts of movement behaviors were categorized into four groups: consistently meet the guideline, from meet to not meet the guideline, from not meet to meet the guideline, and consistently not meet the guideline. We hypothesized that compared with those who kept healthy (consistently meet the guidelines), university students who maintained unhealthy (consistently not meet the guidelines) and changed from being healthy to unhealthy (changed from meet to not meet guidelines) had higher levels of PTSD.

## 2. Materials and Methods

### 2.1. Participants

This longitudinal study was conducted among undergraduates in Guangdong province, China. The recruitment procedure has been described elsewhere [29] and is shown in Figure 1. Briefly, all undergraduates in one university were invited to this study if they were healthy physically and mentally. The questionnaire was generated by an online platform (Wenjuanxing; https://www.wjx.cn/, accessed on 10 December 2021). The students were directed to an online questionnaire by scanning the QR code or clicking the link distributed by staff and student helpers. The first survey was conducted one year after the outbreak of the COVID-19 pandemic, i.e., between 4 December 2020 and 4 January 2021 (Time 1). The second survey was conducted between 15 November and 3 December 2021 (Time 2). Only students who completed both surveys were included in this study. During the data collection period (Time 1, Time 2), the pandemic in China was stable, and fewer than 100 daily new cases were confirmed on most days [32,33]. School closures were relieved, and students’ daily routine returned to normal. Ethics approval was obtained from the Research Ethics Committee, Beijing Normal University & Hong Kong Baptist University United International College (Ref. No.: REC-2021-07).

### 2.2. Exposures: Movement Behaviors

PA and sedentary time were measured using the Chinese version of the International Physical Activity Questionnaire-Short Form (IPAQ-SF), which was reliable (intraclass correlation coefficient > 0.70) and valid among Chinese university students [34]. Moderate-to-vigorous intensity PA (MVPA) was defined as the sum of moderate-intensity PA (MPA) and vigorous-intensity PA (VPA). Sedentary time was abstracted from IPAQ-SF and included all time spent sitting. Regarding screen time, participants reported their average total time spent on TV viewing and computer/tablets/mobile phone use. Sleep duration and sleep quality were assessed using the Chinese version of the Pittsburgh Sleep Quality Index (PSQI), which has shown good reliability (Cronbach’s alpha = 0.85) and validity [35]. Sleep duration was defined as the difference between the bedtime and the wakeup time on the subsequent day. Four options (“0, very good”, “1, fairly good”, “2, fairly bad”, and “3, very bad”) were provided for sleep quality. and the answers were integrated into two categories: good quality (“0” and “1”) and bad quality (“2” and “3”). More details about the data screening of PA, sedentary time, screen time, and sleep can be found in a previous study [29].

The longitudinal shifts of movement behaviors were evaluated based on the Canadian 24-h movement guidelines [31]. Meeting guidelines was defined as: (1) meeting the PA guideline: at least 150 min of MVPA per week; (2) meeting the SB guideline: no more than 8 h of sedentary time and no more than 3 h of screen time; (3) meeting the sleep guideline: 7 to 9 h of sleep per day [31]. Accordingly, the longitudinal shift of movement behaviors (PA, SB, sleep duration) was divided into four categories: consistently meet the guidelines, from meet to not meet the guidelines, from not meet to meet the guidelines, and consistently not meet the guidelines. Correspondingly, the longitudinal shift of sleep quality consisted of four categories: consistently good quality, from good to bad quality, from bad to good quality, and consistently bad quality.

### 2.3. Outcome: Posttraumatic Stress Disorder (PTSD)

The Chinese version of the Posttraumatic Stress Disorder Checklist-Civilian Version (PCL-C) was adopted to measure PTSD. The PCL-C has shown good reliability (Cronbach’s alpha = 0.89) and validity [36]. Based on Diagnostic and Statistical Manual of Mental Disorders, fourth edition (DSM- IV) [37], the PCL-C covers three dimensions (reexperiencing, avoidance, hyperarousal) and consists of 17 items in total. Five options were provided for each item, from “1, not at all” to “5, extremely”. The total score (ranging from 17 to 85) was obtained by summing up all items, with a higher score indicating a more severe symptom. In addition, students were considered having PTSD if they reported having symptoms (i.e., rated from 3 to 5) on at least one item of reexperiencing, three items of avoidance, and two items of hyperarousal.

### 2.4. Covariables: Demographic Information

Demographic information including age, sex, height, and weight were self-reported. Body mass index (BMI; kg/m^2^) was calculated as weight (kg) divided by height (m^2^).

### 2.5. Statistical Analyses

Continuous variables were presented as mean and standard deviation, categorical variables were presented as number of participants and percentages. Paired *t*-tests and Chi-squared tests were used to examine the difference of variables between Time 1 and Time 2. Similar to other studies examining the relationships between movement behaviors and mental health [29,38], demographic factors (age, sex, BMI) were included as covariates. Unadjusted generalized linear mixed models with Gamma distribution were conducted to examine the relationships between movement behavior shifts and PTSD at Time 2 (Model 1). After adjusting for demographic factors and baseline PTSD (Time 1), generalized linear mixed models with Gamma distribution were conducted to predict PTSD at Time 2, with one movement behavior shift (PA, SB, sleep duration, sleep quality) included as a predictor in each model (Model 2). In model 3, generalized linear mixed models with Gamma distribution were conducted, with all demographic factors, baseline PTSD (Time 1), and all movement behavior shifts included. SPSS 27.0 (IBM, Armonk, NY, USA) was used for data analysis, and *p* < 0.05 was considered significant.

## 3. Results

A total of 569 undergraduates completed the survey in both Time 1 and Time 2, and 410 of them (20.2 ± 1.0 years old [Time 2], 41.2% males) provided complete and valid data of demographics, movement behaviors, and PTSD and were included in the final data analysis. Compared with the excluded participants, the included students had a lower proportion of females (58.8% vs. 69.2%, *p* = 0.022), higher MVPA at Time 2 (171.5 ± 271.0 vs. 129.2 ± 206.5 min/week, *p* = 0.047), longer sleep duration at Time 1 (7.1 ± 1.0 vs. 6.7 ± 1.4 h/day, *p* = 0.010), and a higher proportion of good sleep quality at Time 2 (76.6% vs. 67.3%, *p* = 0.023). There was no significant difference in other variables (demographics, movement behaviors, and PTSD). The basic characteristics of participants and differences in variables between Time 1 and Time 2 are presented in Table 1. Characteristics in participants with and without PTSD are presented in Table 1. Longer sleep duration and better sleep quality were observed among participants without PTSD.

The longitudinal shifts in movement behaviors are presented in Table 1. From Time 1 to Time 2, 22.2%, 2.0%, and 45.6% of students consistently met the PA, SB, and sleep guidelines, respectively; 43.2%, 79.0%, 19.3% of students consistently did not meet the PA, SB, and sleep guidelines, respectively. Regarding sleep quality, 69.3% of students consistently had good quality sleep, and 9.3% of them consistently had bad sleep quality. Figure 2 shows the time spent in PA, sedentary time, screen time, and sleep duration in Time 1 and Time 2, based on four categories of each behavior. The prevalence of PTSD among students was 5.4% and 13.7% at Time 1 and Time 2, respectively (see Table 1).

After adjusting for age (Time 2), sex (Time 2), BMI (Time 2), and baseline PTSD (Time 1), students who consistently did not meet the SB guideline (*B* = 0.387; 95% confidence interval [CI]: 0.123, 0.651), or changed from meeting to not meeting the SB guideline (*B* = 0.392; 95% CI: 0.118, 0.665) had higher levels of PTSD at Time 2, compared to those who consistently met the guideline (Table 2, Model 3). Similarly, those who consistently had bad sleep quality (*B* = 0.187; 95% CI: 0.054, 0.319), or changed from good to bad sleep quality (*B* = 0.183; 95% CI: 0.076, 0.290) had higher levels of PTSD, compared to those who consistently kept good sleep quality. Compared to those who consistently met the PA guideline, students consistently failing to meet the PA guideline had lower levels of PTSD (*B* = −0.103; 95% CI: −0.201, −0.004).

## 4. Discussion

This longitudinal study examined the shifts in movement behaviors and their associations with PTSD among university students during the COVID-19 pandemic in China. The two timepoints were one year and two years after the pandemic outbreak when the pandemic was stable. Consistent compliance with the movement behavior guidelines was low especially for SB. The key findings were that, compared to students who did not meet guidelines at both time points or had unfavorable changes (i.e., from meet to not meet guidelines), those who consistently met the SB guideline and maintained good sleep quality had lower levels of PTSD.

A higher prevalence of PTSD among university students was reported two years into the pandemic period (Time 2, 13.7%), whereas the prevalence was 5.4% one year after the pandemic outbreak (Time 1). Another previous study among Chinese undergraduates during the early onset of the pandemic (February 2020) showed that the prevalence was 2.7% [39]. Though the above data were from different samples, PTSD seemed to emerge at the initial stage of the pandemic, and its prevalence became even higher two years after the outbreak of the pandemic. Such a phenomenon can be explained by the delayed onset of the PTSD, which has been frequently reported in different populations [40], including university students in China during the COVID-19 pandemic [41]. Furthermore, a systematic review summarizing 88 studies found that the prevalence of PTSD was 22.6% after pandemics (e.g., sudden acute respiratory syndrome [SARS], COVID-19) across all populations, including healthcare workers, infected cases, and the general public [42]. The above information affirms the long-standing existence and high incidence of PTSD, highlighting the urgency of deploying strategies to reduce mental health burden among university students.

The findings about the association between SB shift and PTSD are consistent with our hypothesis, that is, being less sedentary from Time 1 to Time 2 was associated with less symptoms of PTSD. It is worth noting that the SB guideline in the current study included recommendations for both sedentary time and screen time [31], which may partly explain the extremely low compliance rate (2.0%). Previous evidence consistently observed that higher sedentary time and longer screen time were associated with higher levels of mental health problems [43,44], as well as during the COVID-19 pandemic [45]. To be specific, a longitudinal study among adults found that those with increased screen time after the COVID-19 pandemic outbreak had a worse mental health status, i.e., higher levels of depression, stress, and loneliness [46]. Furthermore, Ellingson et al. found that decreased sedentary time across a one-year period was associated with better mental health (e.g., stress) in young adults [47]. The above evidence indicates that behaviors during a long period may predict mental health and shows the importance of maintaining a low level of sedentary behavior for a long period. However, it should be noted that specific domains of SB were not assessed in this study. A previous study conducted in adults found that depression symptoms were associated with leisure sedentary time, whereas largely non-significantly associated with occupational sedentary time [48].

Regarding sleep, the positive association between consistent good sleep quality and low level of PTSD is consistent with our hypothesis and affirmed the findings from our previous study [25]. It is commonly recognized that sleep quality plays an important role in mental health [24,39]. For example, during the COVID-19 pandemic, negative changes in sleep quality was associated with worse mental health (e.g., depression, anxiety, stress) [49]. Furthermore, it is worth noting that the relationship between sleep quality and mental health can be bidirectional, which has been examined in previous studies. Specifically, a longitudinal cohort study among adolescents reported bidirectional associations between sleep problems (e.g., insomnia) and mental health (e.g., PTSD, depression) [50]. Among the clinical population, sleep disturbance was considered as a chief predictor of mental health disorders; in turn, neurophysiological changes in sleep were often seen in patients with mental illness [51]. It can also be supported by the comparison of sleep quality shift between participants with and without PTSD (Table 1), that is, a larger proportion of participants without PTSD had consistent good sleep quality over time, compared with those having PTSD. However, due to the cross-sectional study design, no cause-and-effect relationship can be revealed in the present study. Furthermore, in future studies, device-based measurement can be adopted to provide more objective information about sleep.

Students who consistently met the PA guideline had higher levels of PTSD compared to those who remained physically inactive. This finding is not consistent with our hypothesis but coincides with the results of our previous study [29]. That is, higher total PA during the outbreak peak (January to March, 2020) was associated with a higher risk of having PTSD after the pandemic outbreak peak (Time 1 in this study) among university students in China [29]. A potential explanation may be the non-linear relationship between PA and mental health, that is, higher or lower PA was associated with poorer mental health, compared with an optimal range of PA (2.5 to 7.5 h per week) [52]. The same pattern was reported by another study measuring PA objectively among adults; that is when daily MVPA participation was higher than 50 min, higher PA was associated with poorer mental health [53]. In addition, interestingly, COVID-19 increased population-level interest in active lifestyle. A previous study based on big data analysis showed that following the COVID-19 pandemic outbreak, the interest on the topic of exercise surged immediately and increased sharply [54]. The potential explanations for the above phenomenon include increased availability of discretionary time, messages recommending PA from media, compensation for reduced PA, and increased health awareness [54]. Furthermore, a previous study examined stress-lead changes of PA among young females aged 18 to 23 years old and observed that more than one-in-five increased PA, and 16.5% became more inactive during emotional stress [55]. Furthermore, people who either increased or decreased PA had higher levels of emotional problems, compared with those whose PA remained unchanged [55]. Similarly, a study conducted during the pandemic observed increased MVPA and total PA among university students after the pandemic outbreak compared with before the pandemic [56]. From a practical standpoint, engaging in PA has been found to be a coping strategy in response to stress in previous studies [57]. For example, a national survey of 36,984 Canadians aged 15 years and above reported that 40% of them used exercise to cope with stress [58]. Therefore, the findings in this study were not surprising. However, it is worth noting that PA was measured using a self-reported questionnaire (i.e., IPAQ) in this study. Though IPAQ has shown its high reliability and validity in many countries [59], the recall bias is inevitable. In addition, the exclusion of participants in the final analysis (27.9%) due to incomplete and unreliable data may also have had an impact on findings.

To the best of our knowledge, this is the first study examining the association between longitudinal changes of movement behaviors and PTSD among university students during the pandemic in China. However, some limitations should be noted. Firstly, self-reported questionnaires were used to measure movement behaviors, which may lead to reporting bias. Secondly, a self-reported questionnaire rather than clinical diagnosis was used to measure PTSD symptoms; also, the questionnaire was developed using DSM- IV criteria rather than the most updated DSM-5. However, the questionnaire has been widely used to evaluate PTSD among different population groups [60,61]. Thirdly, similar to other studies conducted among university students during the pandemic [62,63], a greater proportion of females than males was found in this study. However, no gender difference was found in the regression models. Finally, the generalization of our findings should be cautious, given that the development of the pandemic and the strict measures differed across countries. A nationwide survey in China observed high levels of psychological distress among people at the initial stage of the pandemic, due to the rapid spread of the virus and the unprecedented rigorous public health measures taken [64]. Globally, a systematic review of studies from 33 countries, including China, found that the risk of having severe depression was lower in countries that implemented strict measures earlier, and China is the first country that reacted to the pandemic [65].

## 5. Conclusions

The prevalence of PTSD was high among university students two years after the COVID-19 pandemic outbreak. Maintaining a healthy lifestyle (less sedentary behavior and good-quality sleep) is important to mitigate PTSD and should be considered in future interventions to improve mental health among university students. Besides, awareness and caution should be given to university students with elevated PA levels, as it might be related to a higher PTSD. Coping strategies should be developed by universities and public health systems to support students in having a balanced lifestyle and healthy mental health during the pandemic.

## Figures and Tables

**Figure 1 ijerph-19-13449-f001:**
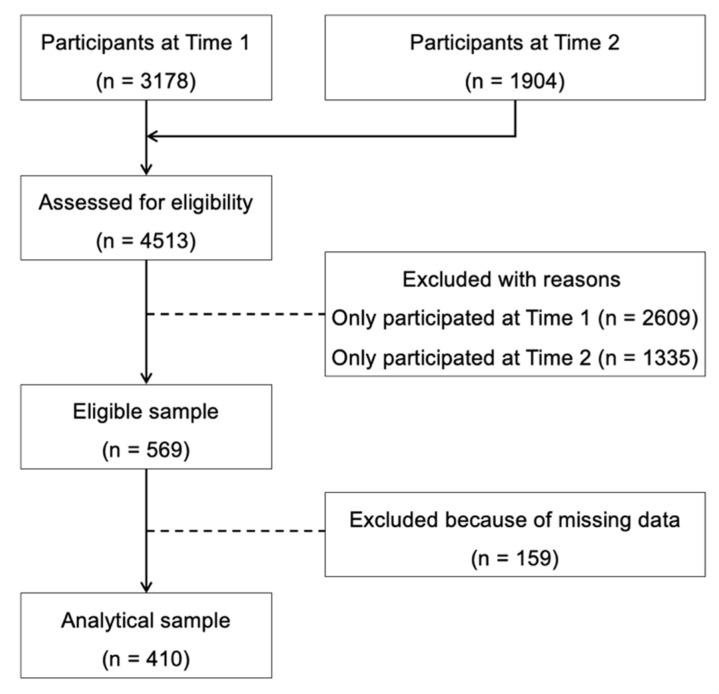
Flow diagram of sample selection. Note. Time 1: between December 2020 and January 2021; Time 2: between November and December 2021.

**Figure 2 ijerph-19-13449-f002:**
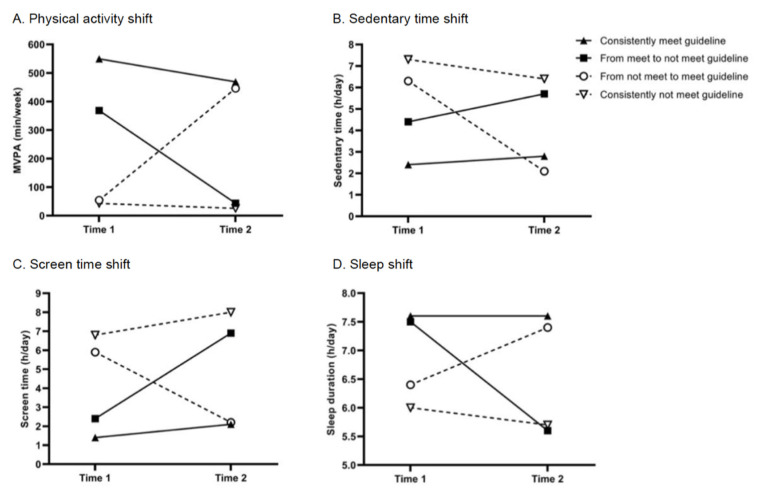
Movement behavior shifts among university students during the COVID-19 pandemic. Abbreviation: MVPA, moderate-to-vigorous intensity physical activity. Note. Time 1: between December 2020 and January 2021; Time 2: between November and December 2021.

**Table 1 ijerph-19-13449-t001:** Characteristics of the longitudinal sample (*n* = 410).

	ALL (*n* = 410)		Mean ± SD or *n* (%)		
	Mean ± SD or *n* (%)	*t*/*χ*^2^ (Time 1 vs. Time 2)	With PTSD at Time 2 (*n* = 56)	Without PTSD at Time 2 (*n* = 354)	*t*/*χ*^2^
Age (year)					
Time 1	19.3 ± 1.0	−69.936 **			
Time 2	20.2 ± 1.0		20.4 ± 1.3	20.2 ± 1.0	1.604
Sex (male)	169 (41.2%)		22 (39.3%)	147 (41.5%)	0.100
Body mass index (kg/m^2^)					
Time 1	22.2 ± 4.8	−1.397			
Time 2	22.5 ± 4.9		22.3 ± 5.2	22.6 ± 4.8	−0.402
MVPA (min/week)					
Time 1	235.9 ± 291.8	4.498 **			
Time 2	171.5 ± 271.0		233.0 ± 396.4	161.7 ± 244.8	1.307
Sedentary time (h/day)					
Time 1	6.7 ± 2.8	2.746 **			
Time 2	6.1 ± 3.0		5.6 ± 2.9	6.2 ± 3.0	−1.176
Screen time (h/day)					
Time 1	6.0 ± 2.8	−9.348 **			
Time 2	7.5 ± 2.7		7.7 ± 2.6	7.5 ± 2.7	0.434
Sleep duration (h/day)					
Time 1	7.1 ± 1.0	2.772 **			
Time 2	6.8 ± 1.5		6.4 ± 1.8	6.9 ± 1.4	−2.107 *
Sleep quality (good)					
Time 1	342 (83.4%)	5.976 *			
Time 2	314 (76.6%)		32 (57.1%)	282 (79.7%)	13.672 **
Physical activity shift					
Consistently meet the guidelines	91 (22.2%)		15 (26.8%)	76 (21.5%)	3.630
From not meet to meet the guidelines	42 (10.2%)		8 (14.3%)	34 (9.6%)	
From meet to not meet the guidelines	100 (24.4%)		15 (26.8%)	85 (24.0%)	
Consistently not meet the guidelines	177 (43.2%)		18 (32.1%)	159 (44.9%)	
Sedentary behavior shift					
Consistently meet the guidelines	8 (2.0%)		0 (0%)	8 (2.3%)	1.956
From not meet to meet the guidelines	13 (3.2%)		1 (1.8%)	12 (3.4%)	
From meet to not meet the guidelines	65 (15.9%)		8 (14.3%)	57 (16.1%)	
Consistently not meet the guidelines	324 (79.0%)		47 (83.9%)	277 (78.2%)	
Sleep duration shift					
Consistently meet the guidelines	187 (45.6%)		19 (33.9%)	168 (47.5%)	3.712
From not meet to meet the guidelines	64 (15.6%)		10 (17.9%)	54 (15.3%)	
From meet to not meet the guidelines	80 (19.5%)		13 (23.2%)	67 (18.9%)	
Consistently not meet the guidelines	79 (19.3%)		14 (25.0%)	65 (18.4%)	
Sleep quality shift					
Consistently good	284 (69.3%)		25 (44.6%)	259 (73.2%)	23.055 **
From bad to good	30 (7.3%)		7 (12.5%)	23 (6.5%)	
From good to bad	58 (14.1%)		11 (19.6%)	47 (13.3%)	
Consistently bad	38 (9.3%)		13 (23.2%)	25 (7.1%)	
PTSD score					
Time 1	23.7 ± 9.1	−7.187 **			
Time 2	28.1 ± 12.5		52.2 ± 8.5	24.3 ± 8.0	23.100 **
Having PTSD					
Time 1	22 (5.4%)	16.378 **			
Time 2	56 (13.7%)				

Abbreviation: MVPA, moderate-to-vigorous intensity physical activity; PTSD, posttraumatic stress disorder. Note. Time 1: between December 2020 and January 2021; Time 2: between November and December 2021. ** *p* < 0.01, * *p* < 0.05.

**Table 2 ijerph-19-13449-t002:** Associations between changes in movement behaviors and post-traumatic stress disorder (95% confidence interval) (*n* = 410).

	Posttraumatic Stress Disorder Score (Time 2)		
	Model 1	Model 2	Model 3
Physical activity shift (reference: consistently meet the guidelines)			
From not meet to meet the guidelines	−0.026 (−0.175, 0.124)	−0.019 (−0.160, 0.121)	−0.036 (−0.173, 0.100)
From meet to not meet the guidelines	0.003 (−0.114, 0.119)	−0.003 (−0.111, 0.106)	−0.028 (−0.134, 0.077)
Consistently not meet the guidelines	−0.044 (−0.148, 0.059)	−0.077 (−0.177, 0.024)	−0.103 (−0.201, −0.004) *
Sedentary behavior shift (reference: consistently meet the guidelines)			
From not meet to meet the guidelines	0.305 (−0.051, 0.660)	0.218 (−0.118, 0.553)	0.222 (−0.107, 0.550)
From meet to not meet the guidelines	0.420 (0.124, 0.716) **	0.383 (0.104, 0.662) **	0.392 (0.118, 0.665) **
Consistently not meet the guidelines	0.484 (0.200, 0.767) **	0.401 (0.133, 0.670) **	0.387 (0.123, 0.651) **
Sleep duration shift (reference: consistently meet the guidelines)			
From not meet to meet the guidelines	0.115 (0.001, 0.230) *	0.055 (−0.054, 0.164)	0.039 (−0.068, 0.145)
From meet to not meet the guidelines	0.100 (−0.005, 0.206)	0.089 (−0.011, 0.188)	0.078 (−0.019, 0.175)
Consistently not meet the guidelines	0.175 (0.069, 0.282) **	0.101 (−0.001, 0.203)	0.068 (−0.033, 0.169)
Sleep quality shift (reference: consistently good)			
From bad to good	0.153 (0.006, 0.300) *	0.095 (−0.047, 0.236)	0.065 (−0.075, 0.204)
From good to bad	0.275 (0.165, 0.385) **	0.201 (0.094, 0.309) **	0.183 (0.076, 0.290) **
Consistently bad	0.314 (0.182, 0.446) **	0.198 (0.066, 0.329) **	0.187 (0.054, 0.319) **

Model 2: adjusted for age (Time 2), sex (Time 2), body mass index (Time 2), and posttraumatic stress disorder score (Time 1). Model 3: included physical activity shift, sedentary behavior shift, sleep duration shift, sleep quality shift, and adjusted for age (Time 2), sex (Time 2), body mass index (Time 2), and posttraumatic stress disorder score (Time 1). Note. Time 1: between December 2020 and January 2021; Time 2: between November and December 2021. ** *p* < 0.01, * *p* < 0.05.

## Data Availability

The data presented in this study are available upon reasonable request from the corresponding author.

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
