# Peer review of "Longitudinal Shifts in Movement Behaviors during the COVID-19 Pandemic: Relations to Posttraumatic Stress Disorder among University Students"

_ijerph, 2022, doi:10.3390/ijerph192013449_

Round 1
Reviewer 1 Report
September 1, 2022
Dear
Ms. Lydia Zeng
Assistant Editor
International Journal of Environmental Research and Public Health
Thank you for the opportunity to review “Longitudinal shifts in movement behaviors during the COVID-19 pandemic: relations to posttraumatic stress disorder among university students”. I found the paper interesting with potential implications. Yet a number of issues should be addressed in order for the manuscript to make a substantial contribution to the literature. Suggested changes are detailed below.
1. One of the main and interesting findings of the study was that students who consistently met the PA guideline had higher levels of PTSD, compared to those maintained physically active. However, it seems that this finding is not mentioned in the abstract. If so, it is suggested to address this finding in the abstract.
2. According to the updated version of the DSM 5, PTSD includes 4 clusters. Thus, when describing PTSD on page 2 line 51-53 it is suggested to mention all 4 clusters.
3. In the introduction when reviewing PTSD and movement behavior the authors generally state that regular and sufficient PA is associated with lower levels of mental health problems. However, in the study mentioned by authors (ref 27) they found that higher PA levels during the outbreak peak were associated with a higher odds ratio of having PTSD and higher levels of re-experiencing and avoidance. Therefore, and considering the study findings it is suggested to elaborate a little more regarding this type of relationship as well.
4. It seems as though PTSD was examined using the DSM IV criteria for PTSD and not the DSM 5 criteria. It is suggested to mention this in the manuscript when describing the questionnaire and in the discussion (and limitations).
5. Regarding the demographic information, it is unclear why body mass was examined (BMI). If there is a reason for gathering this information, please elaborate and explain.
6. Since the study was conducted at two time points, I wonder if the authors examined if there were significant differences between the variables at Time 1 and Time 2 (for example with paired-samples t test). This type of examination could offer additional knowledge and aspects regarding the variables as well as for the discussion.
7. It appears that the variables PA, SB, and sleep were measured on an ordinal scale. For simplicity it might be worth considering (not mandatory) presenting the total average of each variable and showing the changes that occurred in these variables between time 1 and time 2.
8. Regarding the research models it is suggested to elaborate more about the type of statistical analyses that were conducted.
9. In the current study PTSD prevalence as well as the total score of PTSD seems much higher at Time 2 compared to Time 1. This is an important finding that should be addressed with more attention in the discussion (in other words the authors mainly focused on the fact that its prevalence remained high rather than also elaborating that it became higher).
10. One of the most interesting findings of the study was that students that consistently met the PA guideline had higher levels of PTSD, compared to those maintained physically active. Therefore, it is recommended to elaborate a little more regarding this finding and emphasize its uniqueness and importance. For example, it is possible that those with emotional distress were more inclined to PA following professional suggestions for coping with the emotional consequences of COVID. It is also recommended to refer to the practical aspects that can be learned from this finding.
11. Within this context it is worth considering (not mandatory) to add in the conclusion that awareness and caution should also be given to those who demonstrate PA as it might be related to higher PTSD.
Author Response
We would like to thank the reviewers for the constructive comments. The manuscript has now been substantially revised accordingly, and all revisions are marked up using the “Track Changes” function in the revised manuscript.

Reviewer 2 Report
Good manuscript, and this will be appalling to readers. However, I have a few comments. The manuscript especially the introduction section, needs grammar checks (such as punctuation and wording). In the Material and methods section, kindly include the reliability and validity of the scales used to enhance those paragraphs.
Author Response

(The authors gave the same response as above.)

Reviewer 3 Report
The aim of the present study was to investigate whether changes in physical activity behavior during and after the COVID-19 pandemic were associated with PTSD.
Introduction:
__a: It is not yet clear to me why the COVID-19 pandemic is having a negative impact on mental health. Please describe that more clearly. Further, it is not clear why the pandemic can cause PTSD or why the pandemic is a traumatic event? Again, better infer and describe.
__b: Please state the specific objectives
Methods:
__a: The setting of the study and the data collection is not clearly described. What were the eligibility criteria, and the sources and methods of selection of participants? Explain how study size was arrived at. Consider use of a flow-diagram.
__b: Please clearly define all outcome, exposures and covariables. Give also details of methods of assessment.
__c: Why were the Canadian 24-hour movement guidelines used as a reference, and what are the thresholds?
__d: Please specify exactly the variables included in the regression models
Results:
__a: I miss bivariate analyses; so, what is the mean in the group with PTSD compared to the group without?
__b: Please give unadjusted estimates, and make explicit clear which confounders are adjusted for
__c: Please also report the intercept in the linear regression models
__d: Please provide model fit criteria for the ‘Null’ model and the ‘Final’ model
Discussion:
__a: Despite the methodological issues indicated, the authors should mainly strengthen the discussion. The COVID-19 pandemic and the lockdown and hygiene measures in China have directly affected the population under study. It would be advisable to include a reflection on how the results have been influenced by the COVID-19 pandemic and the measures in China.
__b: The authors state that the main finding of their study is that students who meet the SB guideline and maintained good sleep quality had lower levels of PTSD. Is that really the main result of this study? What does compliance with the SB guideline mean? Why was not referred to the WHO Global Action Plan on Physical Activity?
__c: In my opinion, the major weakness of this study is the use of self-reported PA. The authors should address this limitation in more detail.
__d: Please discuss the generalizability of your study results carefully
__e: Lastly, I think there is a lack of specific conclusions related to public health.
Author Response

(The authors gave the same response as above.)

Round 2
Reviewer 1 Report
October 12, 2022
Dear
Ms. Lydia Zeng
Assistant Editor
International Journal of Environmental Research and Public Health
This paper is a revision of the original manuscript entitled “Longitudinal shifts in movement behaviors during the COVID-19 pandemic: relations to posttraumatic stress disorder among university students”.
The manuscript is much improved. The authors addressed the issues that were raised in the original review. The manuscript now reads more clearly: the introduction has been extended, the statistical procedure is more comprehensible, the results section is more precise, and the discussion better integrates the results with the theoretical background and relevant implications.
It was a pleasure reading the revised manuscript.
Author Response
Response: We appreciate the positive feedback from the reviewer.

Reviewer 3 Report
Thank you for the proper revision of the manuscript. Nevertheless, the paper has further weaknesses.
Please revise the flow chart. The total sample is definitly not the sum of participants at T0 and T1; see CONSORT.
Table 1 and Table 2 can be combined.
I am still missing the model fit criteria: AIC, BIC and ICC. Please also report for the null model, i.e. without the predictors to be examined!
Furthermore, several statistical tests have been added. How is this to be handled? Keyword: multiplicity. Please also discuss.
Please also indicate the number of cases considered for the regression models, in Tab.3!
Furthermore, it is not clear to me why not all 3178 participants were included in the analysis?! One strength of mixed models is that cases with missing values can also be included in the analysis! A reduction of the existing sample by 87% (3178 --> 410) is not acceptable. Also, this aspect is not discussed.
Author Response
Response to Reviewer 3 Comments
Thank you for the proper revision of the manuscript. Nevertheless, the paper has further weaknesses.
- Please revise the flow chart. The total sample is definitly not the sum of participants at T0 and T1; see CONSORT.
Response 1: We have now replaced “total sample” with “assessed for eligibility”.
- Table 1 and Table 2 can be combined.
Response 2: We have now combined Table 1 and Table 2.
- I am still missing the model fit criteria: AIC, BIC and ICC. Please also report for the null model, i.e. without the predictors to be examined!
Response 3: We thank the reviewer for this suggestion. According to the model summary of generalized linear mixed models shown in SPSS, Akaike Corrected and Bayesian were used as the criterion. However, the most commonly reported outcomes using generalized linear mixed models in previous studies were Coefficient, 95% confidence interval, and P value, while AIC, BIC or ICC was rarely reported (Guo et al., Br J Sports Med, 2020; Kariippanon et al, Med Sci Sport Exerc, 2022; Xu et al., Int J Environ Res Public Health, 2022). To make the Table concise and readable, the information of criteria is not added in the table.
References:
Guo, C.; Tam, T.; Bo, Y.; Chang, L.; Lao, X.Q.; Thomas, G.N. Habitual Physical Activity, Renal Function and Chronic Kidney Disease: A Cohort Study of Nearly 200 000 Adults. Br. J. Sports Med. 2020, 54, 1225–1230, doi:10.1136/bjsports-2019-100989.
Kariippanon, K.E.; Chong, K.H.; Janssen, X.; Tomaz, S.A.; Ribeiro, E.H.; Munambah, N.; Chan, C.H.; Chathurangana, P.P.; Draper, C.E.; El Hamdouchi, A.; et al. Levels and Correlates of Objectively Measured Sedentary Behavior in Young Children: SUNRISE Study Results from 19 Countries. Med. Sci. Sport. Exerc. 2022, 54, doi:10.1249/MSS.0000000000002886.
Xu, D.; Wang, Y.; Li, M.; Zhao, M.; Yang, Z.; Wang, K. Depressive Symptoms and Ageism among Nursing Home Residents: The Role of Social Support. Int. J. Environ. Res. Public Health 2022, 19, 12105, doi:10.3390/ijerph191912105.
- Furthermore, several statistical tests have been added. How is this to be handled? Keyword: multiplicity. Please also discuss.
Response 4: The newly added statistical tests can provide supportive information in discussion, e.g., the relationship between sleep quality shift and PTSD. We have now added more information in Discussion as follows (Page 11, Lines 287-290):
“It can also be supported by the comparison of sleep quality shift between participants with and without PTSD (Table 1), that is, a larger proportion of participants without PTSD had consistent good sleep quality over time, compared with those having PTSD.”
- Please also indicate the number of cases considered for the regression models, in Tab.3!
Response 5: We have now added the number of cases in Table 2 (Page 9, Lines 216-217).
- Furthermore, it is not clear to me why not all 3178 participants were included in the analysis?! One strength of mixed models is that cases with missing values can also be included in the analysis! A reduction of the existing sample by 87% (3178 --> 410) is not acceptable. Also, this aspect is not discussed.
Response 6: This manuscript aims to explore the relationship between longitudinal changes of movement behaviors and PTSD. As a result, we only include data from a cohort of students who participated in both Time 1 and Time 2 surveys (n = 569), and we have now added this information in Methods (Page 3, Lines 109-110). Missing data from another 159 students were further excluded because of the incomplete information on at least one of the three movement behaviors (PA, SB, sleep) across a 24-hour day. We have now revised Figure 1 (Flow diagram of sample selection) to clarify the sample selection.
The newly added sentence in Methods is as follows:
“Only students who completed both surveys were included in this study.”
